# Ritter-type amination of C(sp$^3$)-H bonds enabled by electrochemistry with SO$_4$$^{2-}$

Ling Zhang[1,4], Youtian Fu[1,4], Yi Shen[1], Chengyu Liu[1], Maolin Sun[2], Ruihua Cheng [2], Weiping Zhu[3], Xuhong Qian[3], Yueyue Ma [2✉] & Jinxing Ye [1,2✉]

By merging electricity with sulfate, the Ritter-type amination of C(sp$^3$)-H bonds is developed in an undivided cell under room temperature. This method features broad substrate generality (71 examples, up to 93% yields), high functional-group compatibility, facile scalability, excellent site-selectivity and mild conditions. Common alkanes and electron-deficient alkylbenzenes are viable substrates. It also provides a straightforward protocol for incorporating C-deuterated acetylamino group into C(sp$^3$)-H sites. Application in the synthesis or modification of pharmaceuticals or their derivatives and gram-scale synthesis demonstrate the practicability of this method. Mechanistic experiments show that sulfate radical anion, formed by electrolysis of sulfate, served as hydrogen atom transfer agent to provide alkyl radical intermediate. This method paves a convenient and flexible pathway for realizing various synthetically useful transformations of C(sp$^3$)-H bonds mediated by sulfate radical anion generated via electrochemistry.

[1] Engineering Research Centre of Pharmaceutical Process Chemistry, Ministry of Education; Shanghai Key Laboratory of New Drug Design, School of Pharmacy, East China University of Science and Technology, 130 Meilong Road, Shanghai 200237, China. [2] School of Biomedical and Pharmaceutical Sciences, Guangdong University of Technology, Guangzhou 510006, China. [3] State Key Laboratory of Bioreactor Engineering, Shanghai Key Laboratory of Chemical Biology, School of Pharmacy, East China University of Science and Technology, Shanghai 200237, China. [4]These authors contributed equally: Ling Zhang, Youtian Fu. ✉email: mayueyue20121@gdut.edu.cn; jinxingye@gdut.edu.cn

Given the high prevalence of amines in natural products, agrochemicals, pharmaceuticals and organic materials[1], tremendous progress has been achieved for the construction of C-N bonds, including reductive amination[2], hydroamination[3], and transition metal catalyzed cross coupling (e.g., Ullmann-Goldberg condensation, Chan–Lam coupling, Buchwald–Hartwig amination, etc)[4–6]. Among the various protocols reported, one of the most desirable strategies is the direct C-H amination in terms of high step- and atom-economy. Comparing with amination of C(sp²)-H bonds, direct incorporation of N-containing moieties into C(sp³)-H bonds is highly desirable but also challenging due to low reactivity and poor selectivity. To circumvent these limitations, many notable strategies were developed, including transition metal catalyzed C(sp³)-H activation[7], C-H insertion catalysis[8], and hydrogen atom transfer (HAT) process[9]. The functionalization of distinct C-H bonds via HAT process often proceeds in highly selective mode, thus avoiding the pre-incorporation of directing moiety into the substrate. Meanwhile, the selectivity and reactivity of HAT process can be rationally tuned by varying hydrogen acceptor and reaction additive. For instance, highly selective amination of C(sp³)-H bonds has been achieved by employing N-fluorobenzenesulfonimide[10,11], N-hydroxyimide derivatives[12–14] and di-tert-butyl peroxide[15–17] as HAT agents. Besides, 1,5-HAT enabled by nitrogen-centered radicals has also served as a robust and practical strategy in selective amination of remote unactivated C(sp³)-H bonds, especially the classic Hofmann–Löffler–Freytag reaction[18–20].

In recent years, the flourishing development of visible light induced photoredox catalysis enables a myriad of elegant C(sp³)-H functionalizations via HAT process under mild conditions. In particular, the decatungstate anion (W; $[W_{10}O_{32}]^{4-}$)[21,22], in excited state, nitrogen radical[23–25], oxygen radical[26–29], and bromine radical[30] generated via photoredox have been employed as the efficient HAT agents for the selective amination of C(sp³)-H bonds. In addition, electrochemistry, which utilizes electron as oxidant or reductant, has accelerated clean, efficient and scalable transformation of C-H bonds into C-N bonds[31–36]. Generally, these processes based on the direct or indirect anodic oxidation of substrates were highly limited to the relatively active C(sp³)-H bonds, including C(sp³)-H bonds adjacent to heteroatoms[37–40], carbonyl[41], allylic[42], and benzylic groups[43–50].

However, C-H amination through HAT process enabled by electrochemistry and electrophotocatalysis[51–54] could enlarge the substrate scope to unactivated alkanes. For instance, the remote inert C(sp³)-H bonds amination was developed to construct pyrrolidines through 1,5-HAT process, which was initiated by the electrochemical generated N radical[55–57]. Using DDQ and Mn(IV) diazide intermediate as HAT agents, Lei group[58] and Ackermann group[59] independently reported manganese-catalyzed photoelectrochemical and electrochemical oxidative azidation of C(sp³)-H bonds, respectively (Fig. 1a).

Several Ritter-type reactions have been disclosed by Baran[60], Liu & Chen[61], providing elegant methods for the amination of inert C(sp³)-H bonds (Fig. 1b). In these works, stoichiometric amounts of expensive oxidants (Selectfluor and hypervalent iodine) were employed to promote the conversion of C(sp³)-H to carbon cation. Lambert and coworkers developed the Ritter type C(sp³)-H amination of benzylic sites[47] and the diamination of vicinal C–H bonds via Ritter-type step[62] by virtue of trisaminocyclopropenium ion (TAC) as the notable electrophotocatalyst (Fig. 1b). König group[29] and Hartwig group[17] reported the reactions of alkanes with simple amides to form the corresponding N-alkyl products using tBuOOtBu as HAT agent (Fig. 1c). In order to enrich the application of Ritter-type C(sp³)-H amination reaction, we conceived to develop an economic manifold that avoids the use of expensive or unstable oxidants, through electrochemical approach.

Generally, thermal, photolysis, electrolysis and metal activated decomposition of peroxodisulfate, which is prepared by the electrolysis of sulfate in industrial process[63,64], can provide a powerful HAT species, sulfate radical anion, thus initiating the C-H bonds functionalization[65–69]. As a consequence, we envisaged that $S_2O_8^{2-}$ could be prepared by the anodic oxidation of $H_2SO_4$ and then would undergo cathodic reduction to yield sulfate radical anion, similar to the protocol reported by Xu group, in which chloride radical was generated by irradiating the anodically formed $Cl_2$ from $Cl^-$ and served as robust HAT agent to form alkyl radical under photoelectrochemical systems[70].

Here, we report a broadly applicable electrochemical strategy for Ritter type C(sp³)-H amination where aliphatic carbon cation is generated via sulfate radical anion mediated HAT followed by further oxidation under electrolysis (Fig. 1d).

## Results and discussion

**Reaction optimization.** Since the product from direct C-H amination of 1,3-dimethyladamantane is used as prodrug for the treatment of Alzheimer's disease, 1,3-dimethyladamantane was chosen as the model reactant to investigate the reaction conditions (Table 1). All the electrolysis reactions were conducted in undivided cells. Based on the conditions screening, the optimal conditions comprised Pt plates as anode and cathode, $H_2SO_4$ as pre-oxidant, $nBu_4NBF_4$ as electrolyte, $CH_3CN$ as solvent and nitrogen source. Under the standard conditions, 93% yield was obtained under 5 mA constant current for 20 h. The reaction did not proceed at room temperature in the absence of electricity (Table 1, entry 2) until the temperature was increased to 65 °C with a low yield of 14% (Table 1, entry 3). Among all the tested acids, $H_2SO_4$ gave the optimal yield. Decreasing the amount of $H_2SO_4$ was detrimental to the reaction yield (Table 1, entries 5–7). Performing the electrolysis without $H_2SO_4$ only gave trace amount of product, demonstrating the pivotal role of $H_2SO_4$ in the C(sp³)-H amination. Notably, the yield was also obviously diminished when the Pt anode was replaced by carbon materials (Table 1, entries 8–10). Either lower or higher current (e.g., 3.0 mA, 7 mA, 10 mA) was not conducive to this reaction (Table 1, entries 11–13). Replacing $H_2SO_4$ with equivalent amounts of $K_2S_2O_8$ could also provide product **1** with 84% yield (Table 1, entry 14). Therefore, peroxodisulfate probably formed by electrolyzing sulfate assisted the rupture of C-H bonds. Besides, the yield could also reach 76% by using $Na_2SO_4$ as sulfate source (Table 1, entry 15). The methanesulfonic acid also showed good performance (Table 1, entry 16) due to the generation of permethanesulfonic acid by the anodic oxidation of methanesulfonic acid, which is similar to peroxydisulfate[71]. Other more detailed conditions were presented in the Supplementary Information (Supplementary Table 1).

**Evaluation of substrate scope.** With the optimized conditions in hand, we next evaluated the substrates scope of alkanes in this protocol (Fig. 2). First, tertiary alkanes were well tolerated and afforded amination products in 44–93% yields with excellent regioselectivity (Fig. 2, 1–8). Of note, the reaction with deuterated acetonitrile as nitrogen source gave C-deuterated acetamination products **2** and **6** with 85 and 78% yields, respectively. Generally, the amination of secondary C(sp³)-H bond is more challenging than that of tertiary C(sp³)-H bond, so 16.0 equivalents of $H_2SO_4$ is required. A variety of cyclic, secondary hydrocarbons were smoothly transformed into aliphatic amide products (Fig. 2, **9–12**) in moderate to good yields, while the reaction with cyclododecane furnished the product **13** in 32% yield due to the poor solubility in acetonitrile. However, when using different types of linear alkanes, two isomers were obtained with different ratios. The electrolysis of n-hexane provided 1:1 ratio of isomers

**(a) Examples of the azidation of unactivated alkanes**

**(a) Lei , ref.58 : Mn cat.**
**DDQ electrophotocatalyst**
*i*= 8 mA, blue LEDs

**(b) Ackermann , ref.59 : Mn cat.**
*i*= 8.0 mA

**(b) Examples of Ritter-type C-H amination**

**Baran ref.60 : Selectfluor, CuBr₂, Zn(OTf)₂**

**Liu and Chen ref.61 : PFBI-OH**
Ru photocatalyst, CFL

**Lambert ref.47,62 : CFL, CH₃CN/TFA**
**TAC electrophotocatalyst**

**(c) Examples of intermolecular amidation of unactivated alkanes**

**Hartwig, ref.17,** Cu cat. *t*BuOO*t*Bu, 100°C

**König, ref.29,** Cu cat. *t*BuOO*t*Bu, LED

**(d) This work**

Pt (+)-Pt (-), H₂SO₄
CH₃CN, r.t.

R¹, R²= alkyl or H
R³= alkyl or aryl

71 examples
up to 93% yields

- Electrochemical formed SO₄·⁻ as HAT agent
- SO₄²⁻ mediated amination, avoiding expensive or unstable oxidants
- Amination of aliphatic C(sp³)-H and benzylic C(sp³)-H
- Broad functional groups tolerance and gram-scale sythesis

**Fig. 1 Examples of C(sp³)-H amination reactions. a** Examples of the azide reaction of unactivated alkanes via (photo)electrocatalysis. **b** Examples of Ritter-type C-H amination via Selectfluor, hypervalent iodine reagent and TAC electrophotocatalyst. **c** Examples of intermolecular amidation of unactivated alkanes via *t*BuOO*t*Bu and Cu cat. **d** This work: Ritter-Type amination of C(sp³)-H bonds enabled by the electrochemistry with SO₄²⁻.

---

**Table 1 Optimization of reaction conditions.**

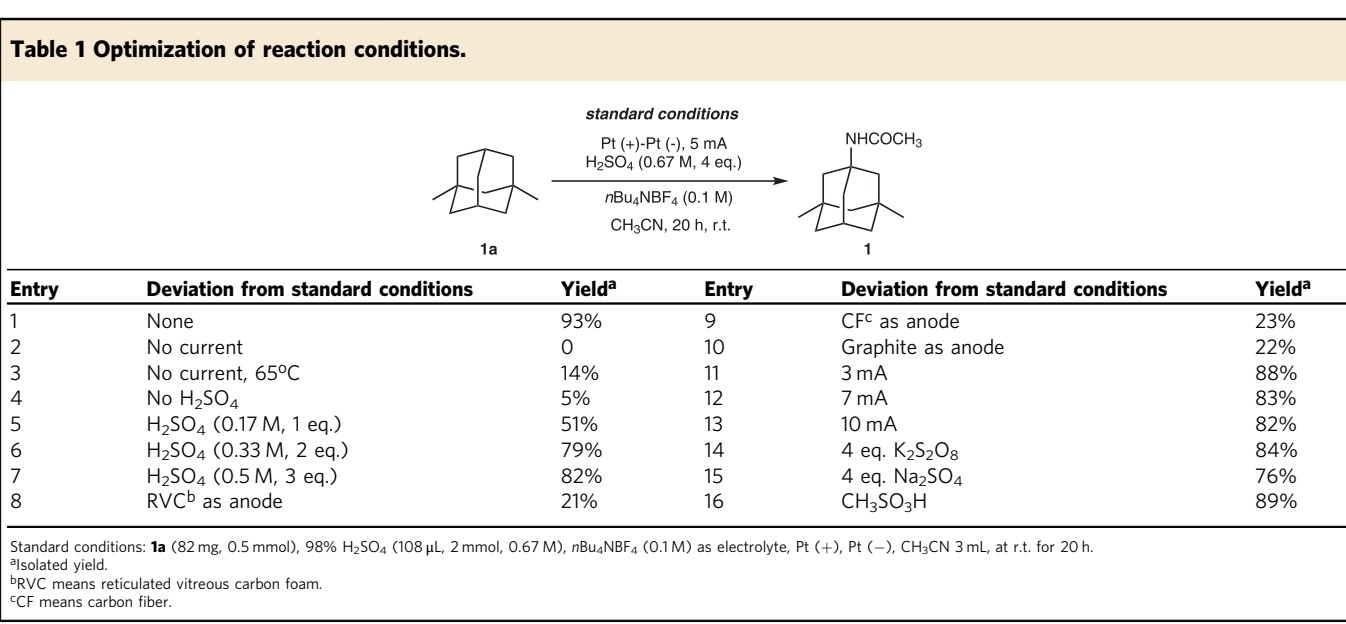

*standard conditions*
Pt (+)-Pt (-), 5 mA
H₂SO₄ (0.67 M, 4 eq.)

*n*Bu₄NBF₄ (0.1 M)
CH₃CN, 20 h, r.t.

**1a** → **1**

| Entry | Deviation from standard conditions | Yield[a] | Entry | Deviation from standard conditions | Yield[a] |
|---|---|---|---|---|---|
| 1 | None | 93% | 9 | CF[c] as anode | 23% |
| 2 | No current | 0 | 10 | Graphite as anode | 22% |
| 3 | No current, 65°C | 14% | 11 | 3 mA | 88% |
| 4 | No H₂SO₄ | 5% | 12 | 7 mA | 83% |
| 5 | H₂SO₄ (0.17 M, 1 eq.) | 51% | 13 | 10 mA | 82% |
| 6 | H₂SO₄ (0.33 M, 2 eq.) | 79% | 14 | 4 eq. K₂S₂O₈ | 84% |
| 7 | H₂SO₄ (0.5 M, 3 eq.) | 82% | 15 | 4 eq. Na₂SO₄ | 76% |
| 8 | RVC[b] as anode | 21% | 16 | CH₃SO₃H | 89% |

Standard conditions: **1a** (82 mg, 0.5 mmol), 98% H₂SO₄ (108 μL, 2 mmol, 0.67 M), *n*Bu₄NBF₄ (0.1 M) as electrolyte, Pt (+), Pt (−), CH₃CN 3 mL, at r.t. for 20 h.
[a]Isolated yield.
[b]RVC means reticulated vitreous carbon foam.
[c]CF means carbon fiber.

---

with 54% yield. For linear alkanes containing electron-deficient moieties, such as ester, bromine, and chlorine, the aminations were apt to take place at the secondary C-H bonds far away from these functional groups, leading to the mixture of acetamination products **17–19** with different ratios about 2.3:1, 1.5:1, and 1.3:1, respectively. To our delight, the amination of 1-bromopentane and 1-chloropentane occurred preferentially at C-H bond away from halogens with high regioselectivity and provided single products in good yields (Fig. 2, **15**, **16**).

Next, the scope of alkylbenzenes was assessed (Fig. 3). The amination reaction of alkylbenzenes bearing different functional groups at aliphatic chain, such as carbonyl, alkenyl and cyano group, underwent smoothly to deliver **20–30** in 41–76% yields, with less amount of H₂SO₄ and shorter reaction times. Product **24** was afforded with excellent regioselectivity, illustrating that C(sp³)-H amination preferentially reacted at benzylic over tertiary C-H bond. Besides, the phenylpropene underwent isomerization to form **28** in 41% yield. Furthermore, derivatives of toluene and ethylbenzene containing various functional groups on the benzene ring, including halogen, cyano, ester, carbonyl, nitro, amino, methoxyl, and alkyl groups, were well accommodated (Fig. 3, **31–41**). Mono-acetamination product

were provided for the alkylbenzenes with multiple benzylic sites. Commonly, nitro group and iodine atom are prone to be reduced under electrolysis[72–75], but remained intact under our reaction conditions. In particular, this reaction system was tolerated to unprotected amine groups, furnished products **39** and **41** in 25 and 54% yields, respectively, likely as a result of the acidification of amines. It is worth noting that previous electrochemical methods did not work for the amination of alkylbenzenes bearing strong electron-withdrawing groups[43–46,48–50], which are viable reactants under our reaction systems. In addition, the α-amino acid precursors and β-amino acid precursors (Fig. 3, **58–62**) were also obtained in good yields.

With the replacement of acetonitrile by ethyl cyanoacetate, butyronitrile, isobutyronitrile and adiponitrile, the reaction could also take place to yield products **63–68** in moderate to good yields. (Fig. 4a) To further explore the synthetic utility of the methodology, the modification of natural products and pharmaceuticals (Fig. 4b) and scaled-up experiments (see the Supplementary Methods for details) were carried out. The amination of esterified ibuprofen was site-selectively occured at the less hindered benzyl position in 77% yield. The natural deoxyfenisin and fragrance celestolide were funtionalized to give product **70** and

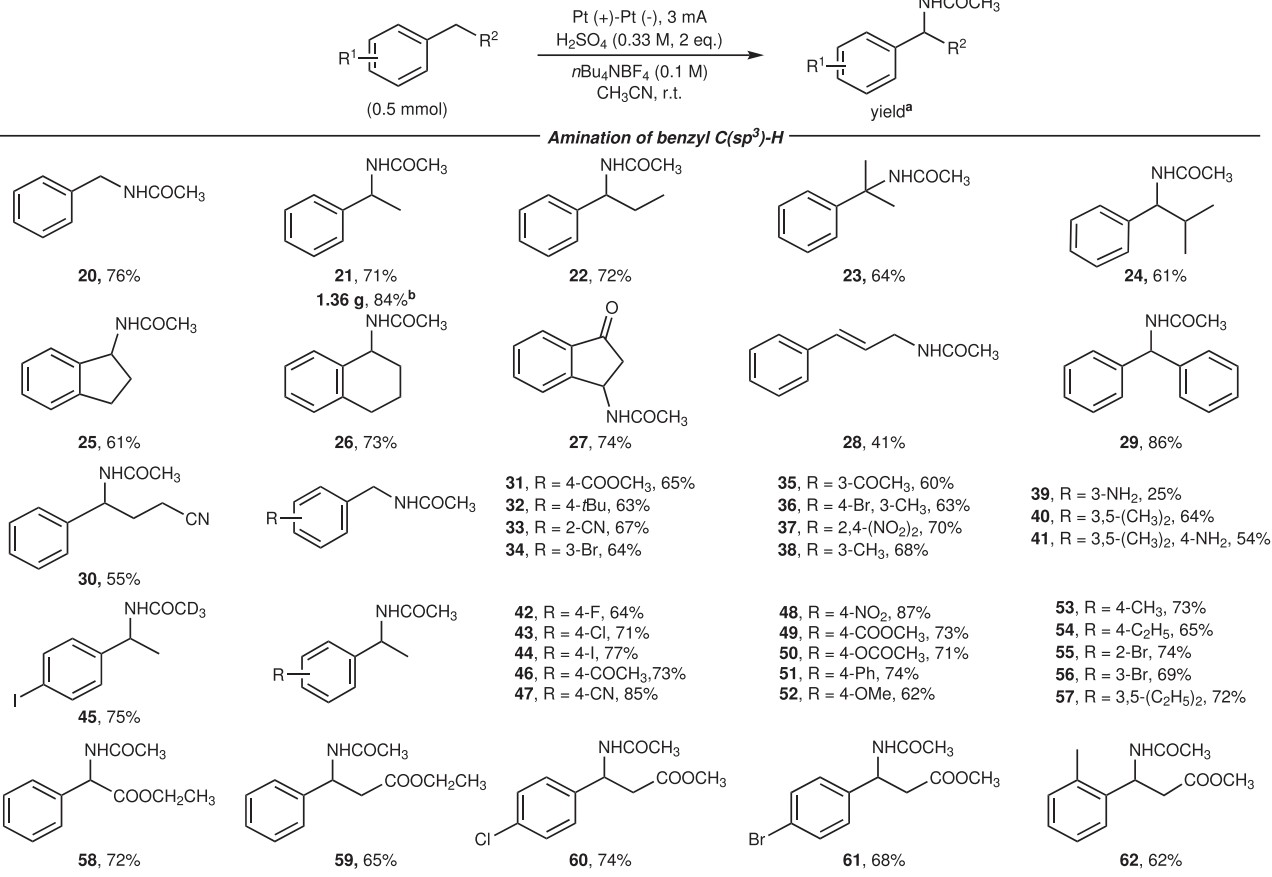

**Fig. 2 Reaction scope of alkyl substrates.** [a]Isolated yield; [b]Electrolysis performed on 10 mmol scale; [c]Regio isomeric ratios were determined by [1]H NMR.

**Fig. 3 Reaction scope of arene-containing substrates.** [a]Isolated yield; [b]Electrolysis performed on 10 mmol scale.

**71**, respectively. Next, we carried out gram-scale amplification experiments, in which aminated products (**1**, **12**, **21**) were obtained with satisfying yields.

Compounds **1b** and **3b** are the precursors of the drug memantine hydrochloride for the treatment of Alzheimer's disease and the antiviral drug amantadine hydrochloride, respectively. Therefore, further hydrolysis of compounds **1** and **3** were shown in Fig. 5. Our methodology could be applied for preparing the precursors of memantine hydrochloride **1b** and amantadine hydrochloride **3b** in two steps from simple materials, avoiding

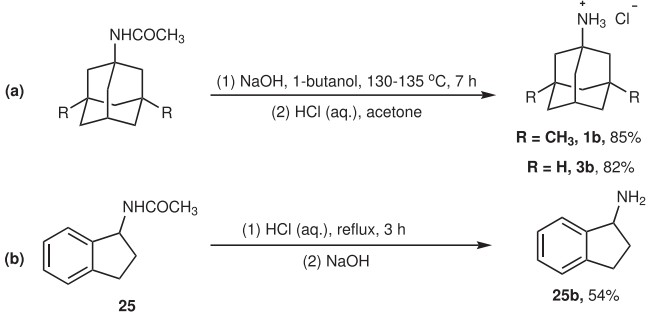

**Fig. 4 Reaction scope other nitriles and the modification of natural products and drug molecules. a** Products of other nitriles. **b** Products of natural products & drugs. Conditions: substrate (0.5 mmol), $H_2SO_4$ (54 μL, 1 mmol, 0.33 M), $n$Bu$_4$NBF$_4$ (0.1 M), $CH_3CN$ (3 mL), 5 mA; [a]Conditions: substrate (0.5 mmol), $H_2SO_4$ (108 μL, 2 mmol, 0.67 M), $n$Bu$_4$NBF$_4$ (0.1 M), $CH_3CN$ (3 mL), 5 mA.

**Fig. 5 Hydrolysis of amide products. a** Conditions: Amide compound (3.0 mmol, 1 eq.), NaOH (24 mmol, 8 eq.), 1-butanol as solvent, 130–135 °C for 7 h. Then, the free base was added 5 M aq. HCl (3 mL) to give hydrochloride. **b** Conditions: **25** (3.0 mmol, 1 eq.), HCl (3 M, 10 eq.), reflux for 3 h.

the use of bromine and the large amount of $H_2SO_4$ while maintaining high yields (79 and 57% total yields). The hydrolysis of **25** also yielded **25b**, a key intermediate for the synthesis of rasagiline(see the Supplementary Methods for details).

**Mechanistic studies**. To obtain more insights into the Ritter-type amination reaction mechanism, a series of mechanistic studies were conducted. First, replacing sulfuric acid with sodium sulfate, potassium peroxodisulfate and methanesulfonic acid, which are unlikely to facilitate the Ritter-type step, 76–89% yields of desired product were provided (Fig. 6a). Besides, the formation of persulfate via the electrolysis of sulfuric acid under standard conditions was also proved by UV absorption spectra experiment (Supplementary Fig. 6). So, sulfuric acid was proved to promote the activation of $C(sp^3)$-H rather than as an additive to facilitate the Ritter-type step. 65% yield of product **1** was delivered by electrolyzing **1a** at anodic chamber under standard conditions, demonstrating the generation of sulfate radical cation through the direct anodic oxidation (Fig. 6a).

By comparing the oxidation potentials of sulfate ($E^o$ = 2.6 V vs NHE) and some alkanes in literatures[76–80], sulfates are likely to be oxidized by anode in prior to alkanes. However, heating or irradiating $K_2S_2O_8$ with **1a** did not afford the product **1** (Fig. 6a). We envisaged that the conversion of carbon radical to carbon cation could be accomplished by anode oxidation rather than the single electron transfer of $SO_4^{-\cdot}$. Radical clock experiment confirmed that radical intermediate **72b** was generated and underwent ring opening and further oxidation, furnishing carbon cation intermediate (Fig. 6b).

Based on the studies mentioned above, a possible mechanism for this reaction process was shown in Fig. 6c. Intially, sulfate anion $SO_4^{2-}$ underwent a single electron oxidation process at the anode to afford $SO_4^{-\cdot}$. As a powerful HAT agent, $SO_4^{-\cdot}$ abstracted hydrogen atom from the substrate **1a** to generate carbon radical **73**, which was oxidized at the anode to the carbocation intermediate **74**. Subsequently, nucleophilic attack of acetonitrile to **74** through classic Ritter reaction process provided final product **1**. At the same time, the dimerization of $SO_4^{-\cdot}$ generated $S_2O_8^{2-}$, which was then transferred to the cathode area and reduced back to $SO_4^{-\cdot}$.

In summary, we have developed an electrochemical approach that can be applied for the Ritter-type amination reaction of aliphatic $C(sp^3)$-H and benzylic $C(sp^3)$-H under mild conditions. By electrolyzing sulfate, $SO_4^{-\cdot}$ was generated and abstracted hydrogen atom from $C(sp^3)$-H bonds. This electrochemical method also exhibited excellent regioselectivity and functional groups tolerance, rendering the reaction simple, safe, widely applicable. It is anticipated that this method would provide a convenient and alternative entry for the synthesis of amines derivatives.

## Methods
**Representative procedure for amination**. An oven-dried undivided cell was equipped with a stir bar, 1,3-dimethyladamantane (82 mg, 0.5 mmol, 1 eq.), 98% $H_2SO_4$ (108 μL, 2 mmol, 0.67 M, 4 eq.), $n$Bu$_4$NBF$_4$ (98 mg, 0.3 mmol, 0.1 M), $CH_3CN$ (3 mL). Air has little effect on this reaction. The assembled electrodes were placed into the solution. The silica gel plug was sealed with film. The mixture was

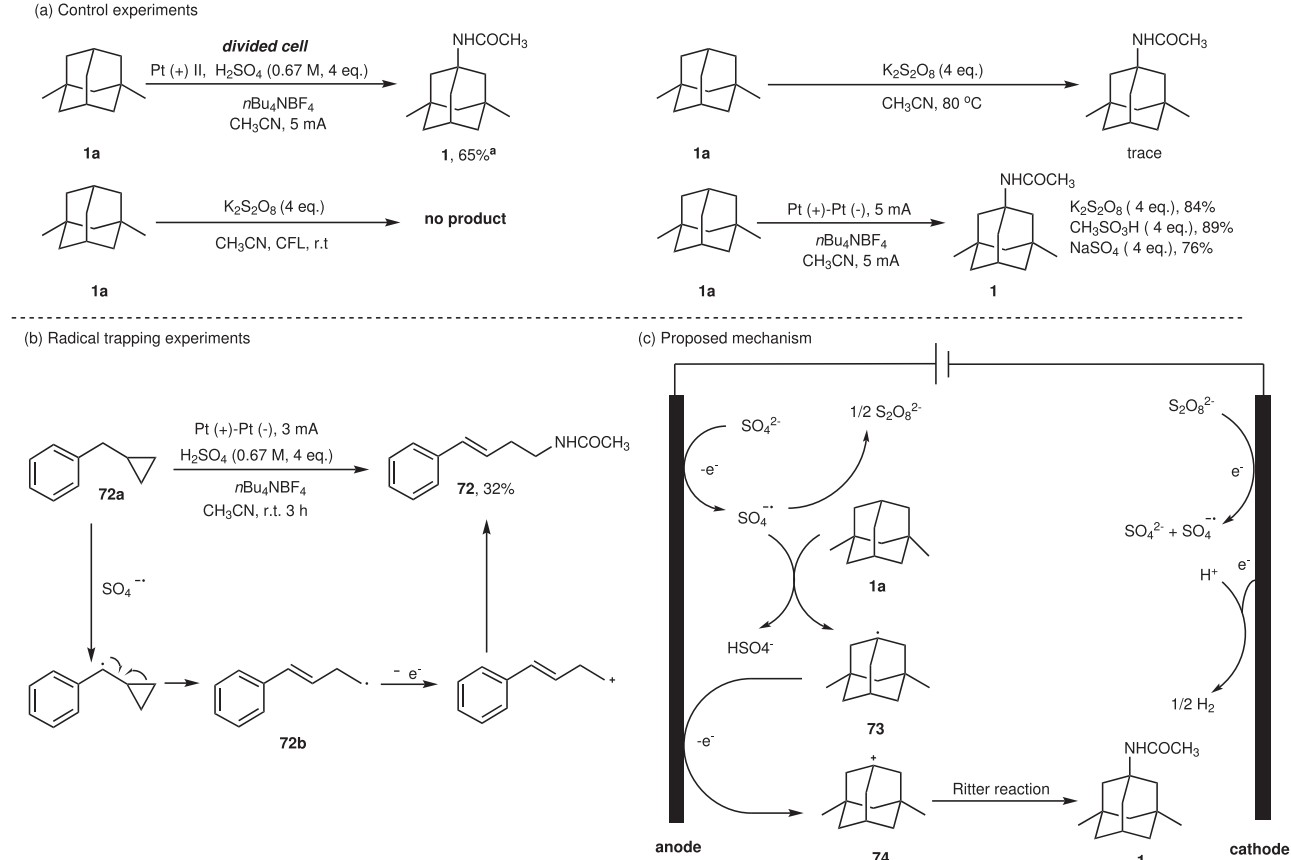

**Fig. 6 Mechanistic studies. a** Control experiments. [a]For the reaction, to the anodic chamber was added **1a** (0.5 mmol), $H_2SO_4$ (108 μL, 2 mmol, 0.67 M), $nBu_4NBF_4$ (0.1 M) and $CH_3CN$ (3 mL), the cathodic chamber was added $nBu_4NBF_4$ (0.1 M) and $CH_3CN$ (3 mL). **b** Radical trapping experiments. **c** Proposed mechanism.

electrolyzed at a constant current of 5 mA until the tertiary alkane was completely consumed (See Supplementary Fig. 3). The Pt electrodes were washed by water, ethanol, and DCM in turn. After the reaction is over, drop $NaHCO_3$ saturated solution into the reaction system slowly until no bubbles were observed. The aqueous layer was separated and extracted with EtOAc ($3 \times 10$ mL), and the combined organic layers were washed with brine and dried over anhydrous $Na_2SO_4$. Following concentration in vacuo, the crude product was purified by column chromatography on silica gel to give pure product **1** (103 mg, 93%; see the Supplementary Methods for details).

## Data availability

The authors declare that the data supporting the findings of this study are available within the article and its Supplementary Information file. For experimental details and compound characterization data see Supplementary Methods. For [1]H NMR, [13]C NMR, and [19]F NMR spectra see Supplementary Figs. 7–82.

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

## Acknowledgements

This work was partially sponsored by the National Natural Science Foundation of China (22071056, J.Y.; 22001070, Y.M.), and Shanghai Sailing Program (20YF1410400, Y.M.).

## Author contributions

L.Z. and Y.F. contributed equally to this work. L.Z., Y.F., and Y.S. performed the experiments, analyzed the data. C.L. and M.S. assisted the purification of compounds and analysis of data. R.C., W.Z., and X.Q. contributed to mechanistic studies. L.Z., Y.F., Y.M., and J.Y. wrote the manuscript, supplementary methods, and related materials. Y.M. and J.Y. designed and directed the project. L.Z., Y.F., R.C., W.Z., X.Q., Y.M., and J.Y. revised the manuscript.

## Competing interests

The authors declare no competing interests.
