## [Peer Review File · Nature Communications]

REVIEWER COMMENTS

Reviewer #1 (Remarks to the Author):

Authors report a sulfuric acid-enabled photoelectrochemical Ritter-type amination of benzylic and unactivated C-H bonds. Perhaps the most relevant prior art is that reported by Lambert and co-workers (Ref. 17) whom reported an electrophotocatalytic Ritter-type amination but substrate scope was limited to alkylbenzenes. The tolerance of unactivated C(sp³)-H bonds (eg adamantane) in this work is an impressive improvement. The reaction is versatile in scope and with high selectivity, can be conducted up to gram scale. The applicability to late-stage functionalization has been demonstrated by amination of several nature products and drug molecules (Scheme 4c). Considering the hot emerging topic of synthetic photoelectrochemistry and the key ability to involve unactivated Csp³-H bonds in a broad scope, my view is that this manuscript in principle carries the novelty and broad interest for publication in Nat. Comm.

However, there are some serious concerns that need to be addressed before I can recommend acceptance. Particular points of concern are a notable lack of citations of key related work and the validity of claims that reactions are "photoelectrochemical". To clarify, whether the reaction is photoelectrochemical or electrochemical is not a dealbreaker for acceptance, but readers deserve to know which via thorough validation of claims not via supposition.

Major experimental revisions:

1. As authors note the reaction does proceed efficiently without light (Table 1 entries 1 (97%) vs 4 (74%), and Fig. 1A) and homolysis of persulfate is known at temperatures above rt (e.g. 40-60 oC). Elsewhere, increased reaction temperature as a result of hi power LED irradiation is known to promote photochemical reaction yields: ChemPhotoChem 2021, DOI:10.1002/cptc.202100059. Therefore, can this ca. +20% yield over 12-20 h rxn time simply be a thermal effect from LED heat on the homolysis of persulfate? (Same question over signal formation in EPR with LED). Of course the answer to this key above question influence the whole manuscript, including title. Authors must conduct reaction at a controlled, low temperature (e.g. 0 oC) to check the difference with vs without light. e.g. SI entry 8 at 10 oC (81%) should be checked without light. Authors should measure the internal temperature of a reaction under optimal conditions.

2. Persulfate is a white solid and does not absorb visible light in MeCN; its absorption is <400 nm.

It is possible a charge transfer complex between substrate and persulfate may operate to afford a visible absorption. Authors should report the UV-vis spectra of their reaction mixtures and components.

3. The claim results in Scheme 4A are photoelectrochemical is certainly a red flag. There is no obvious driving force for that C-O cleavage - how does the mechanism work in these cases? Classic acid-promoted SN1 Ritter reaction of activated alcohols using sulfuric acid is known even at rt, and cannot be ruled out. The reaction of tert-butanol is a textbook reaction students learn for the SN1 Ritter hydrolysis. Hydrolysis of ethers is also known to be achieved with strong acids.

Authors must examine the control reactions (i.e. no light, no current, without both light+current) with adamantanol to check the efficiency. If it is confirmed to be simply a classic Ritter reaction, Scheme 4A's results can be omitted and this does not detract impact or hinder potential acceptance, in my view.

4. Scheme 4A: Authors should check the reactions of 2x 4-substituted phenols (1x Elec-withdrawing group and 1x Elec-donating group), to check if their C-O cleavage reactivity works for C(sp²)-OH.

Major text revisions:

1. sp² C-H amination is also reported in scheme 3. However, key related work on photoelectrochemical sp² C-H aminations, especially those of electron-deficient arenes, are missing in the citations and must be cited: *Angew. Chem. Int. Ed.* 2021, 60, 11163; *Org. Chem. Front.* 2021, 8, 1132; *ChemRxiv* 2021, DOI: 10.26434/chemrxiv.13718392.v1; *ChemElectroChem* 2021, 8, 1571. The authors's ms shows a key advantage over these reports in terms of simpler reaction conditions and shorter reaction times so this is useful for readers.

2. Xu group recently reported a HCl enabled photoelectrochemical Minisci-type reaction. Anodic generated Cl₂ undergoes homolysis with vis light irradiation affording chlorine radicals which are powerful HAT agents that can activate also unactivated hydrocarbons (*Angew. Chem. Int. Ed.* 2020, 59, 14275). This seminal work, which also uses photoelectrochemistry with a simple mineral acid, is very related in concept to the current manuscript in mechanism. It is unfortunately not cited - it must be cited and described clearly in the introduction.

3. Related to Scheme 4A: Authors should mention in the text that photoelectrochemical C-O (sp³/sp²) cleavage reactions are previously reported, but so far only under reductive conditions.

These reports should be cited: *Angew. Chem. Int. Ed.* 2021, DOI:10.1002/anie.202105895 and *Angew. Chem. Int. Ed.* 2021, DOI: 10.1002/anie.202107169.

Minor revisions:

1. The authors mention "Up to now, in the field of photoelectrochemistry, the conversion of C(sp³)-H bonds to C-N bonds is usually constructed by coupling reaction of intramolecular amination." This is confusing, since up to now the conversion of C(sp³)-H bonds to C-N bonds are only by oxidation and then Ritter-type reaction (Ref 17) or hydrogen atom transfer (*JACS* 2020, 142, 1698 and *JACS* 2020, 142, 17693). These references should be cited and the description rephrased. This reviewer agrees that Refs 15-16 contain some related works, but many of them either i) not using photoelectrochemical methods or ii) are not C(sp³)-H aminations and they should be omitted or cited elsewhere in the manuscript.

2. Authors describe oxidative ability of sulfate radical, but sulfate radical is also a well-known HAT agent (Refs 19-21). Since the mechanism of unactivated Csp³-H bonds involves a HAT process, photocatalytic/photochemical persulfate/sulfate based HAT reactions should be briefly mentioned and relevant examples cited: *Chem. Sci.* 2016, 7, 2111-2118; *Org. Lett.* 2021, 23, 2002.

3. Xu group recently reported a HCl enabled photoelectrochemical Minisci-type reaction. Anodic generated Cl₂ undergoes homolysis with visible light irradiation affording chlorine radicals which are powerful HAT agents that can activate also unactivated hydrocarbons (*Angew. Chem. Int. Ed.* 2020, 59, 14275). This seminal work, which also uses photoelectrochemistry with a simple mineral acid, is very related in concept to the current manuscript in mechanism. It is unfortunately not cited - it must be cited and described clearly in the introduction.

4. Related to Scheme 4A: Authors should mention in the text that photoelectrochemical C-O (sp³/sp²) cleavage reactions are previously reported, but so far only under reductive conditions. These reports should be cited: *Angew. Chem. Int. Ed.* 2021, DOI:10.1002/anie.202105895 and *Angew. Chem. Int. Ed.* 2021, DOI: 10.1002/anie.202107169.

5. Product 71 should be in Scheme 2, since the starting material is not alcohol or ether.

6. pg4: "equivuivalents" should be equivalents.

7. Table 1: conversion is shown instead of yield. Conversion only means how much starting material is consumed. Authors should show product yield.

8. Ref. 21's author names are incorrect, first names are given full and second name initials. It should be the other way around: Davide, R.  Ravelli, D. etc

9. Table 1: please define "CC" and "CF" (entries 11-12) since these electrode materials are currently unclear

10. Fig. 1: "shed light" : advise to rephrase this to avoid repetition of "light" and possible misunderstandings:

e.g. "in order to unravel/probe the reaction mechanism.."

11. Abstract+pg.2+throughout: Homogenised (scientifically incorrect, different meaning)  homolysed (correct)

12. "deprived a H atom" (could be confused with deprotonation)  abstracted a H atom/underwent hydrogen atom transfer/HAT (clearer)

13. Scheme 4b: show expanded structures of 72-78 to improve readability.

14. Scheme 3: 49 vs 50, 58, 59-62: different depictions of ester structures. Please use structure 49's representation for all.

15. Some substrates are similar and should be grouped by one drawn structure with an R group to simplify, e.g. 31 vs 49 (almost same), 34 vs 36 and 21 vs 22. Readers will not get value from seeing all the different R group structures drawn individually.

Reviewer #2 (Remarks to the Author):

The present work by Ma, Ye and co-workers describes a Ritter-type amination protocol triggered by photoelectrochemical conditions. Unfortunately, the quality of the ms is rather poor, with several pitfalls, and rejection is therefore recommended. On one side, the language/style prevents to actually follow the reasoning by the Authors in several parts. On the other one, let apart the high quality standards required for publication in NC, there are several inconsistencies in the scientific content, that render this contribution too premature to be considered for publication in any journal. The main points supporting the above-mentioned recommendation are reported below.

- Introduction: the Authors focus the attention on the functionalization of C-H bonds, but then their examples include the functionalization of different bond types (C-C via decarboxylation and C-halogen via dehalogenation, C-O bonds in alcohols/ethers). This inconsistency is rather confusing and should be addressed. I would personally remove examples not dealing with C-H bonds, also to be consistent with the mechanism. Furthermore, there is a huge difference between aliphatic vs aromatic C-H bonds (see the case of benzene) in terms of mechanism of activation (aromatic C-H bonds cannot be activated via C-H cleavage!). What about the use of ethers/alcohols? These are not even mentioned, neither in the Introduction, nor in the mechanism.

- Introduction/2: a suitable scheme describing literature precedents is lacking.

- Chemistry: I am firmly convinced that the claim by the Authors, in particular the actual "photoelectrochemical" nature of their protocol, is not justified by their experimental data. Indeed, the data gathered in Table 1 (see entry 4) clearly demonstrate that light is not essential for the process to occur. 74% product yield cannot be considered a minor pathway and the system works nicely also under purely electrochemical conditions. This aspect is not even mentioned by the Authors!

- Figure 1B: equation a in part B should be justified by experimental evidences, not only by literature precedents. On the other hand, equation b in part B is rather confusing because the Authors should actually justify their claims:

* light: is persulfate absorbing light in the blue region?

* electricity: this activation manifold cannot lead to the formation of 2 equiv of sulfate radical anion;

* heat: the process occurs at 25°C, is persulfate homolyzed (not homogenized) at this temperature?

Also, the shown interaction of the persulfate radical anion with the substrate is not representative at all of the diverse interactions reported in the text (see the sub  sub.+ conversion).

- Figure 1C: The Authors should better comment on the shape of the observed EPR signal. It seems that more than a single contribution is present. Also, the parameters of the signal are not consistent with the literature, see DOI: 10.3109/10715769209083142.

Reviewer #3 (Remarks to the Author):

In this work, Zhang and coworkers reported an amination method of unactivated C-H bond and benzylic C-H bond via Ritter type reaction as well as deoxygenative amination of ether and alcohol. This method has a good substrate scope, simple conditions and moderate to high yields. In this study, persulfate salts are in-situ generated by electrochemistry which oxidize C(sp³)-H bond. Same reactivity has been reported in other works which focused on oxygenation (Org. Lett. 2016, 18, 1234–1237; Org. Lett. 2017, 19, 572–575) and azidation (Org. Chem. Front., 2016, 3, 1326–1330) reactions. The direct amination using persulfate as oxidant is novel here since previous Ritter type C(sp³)-H amination requires expensive or unstable oxidants (Selectfluor in J. Am. Chem. Soc. 2012, 134, 5, 2547–2550; hypervalent iodine compounds in Chem. Sci., 2017, 8, 7180-7185 and Chem. Commun. 2016, 52, 13082–13085). The result from this paper is very interesting. However, the author should try to improve the scholar presentation. In some cases, linguistic errors make the paper a bit hard to read. More background research and reference are also needed.

- The term photoelectrochemistry is misused. Photoelectrochemistry is commonly used to describe the use of photoelectrodes or photovoltaics for electrolysis. This work falls into the range of electrophotochemistry. This definition is made in reference 8 of the manuscript.
- Have the authors done a control experiment using a persulfate salt as the stoichiometric oxidant under photoirradiation? This should also provide the desired product based on the proposed mechanism.
- In abstract line 3, “Then S₂O₈²⁻ was homogenized under light to give sulfate radical anion, which deprived hydrogen on the substrate to generate radical intermediate.” Does the author have a typo mistake here? It should be homolyzed rather than homogenized. Same mistake was found in page 9 and 10. In addition, the verb deprive is recommended to be replaced by abstract.
- In abstract line 5, remove with in featured with.
- In abstract line 7, change challenge into challenging.
- Rewriting introduction is recommended. There are many methods to construct C-N bond. Some of the important reactions (reductive amination, Ullman coupling) has not been mentioned. It is recommended to focus on the C(sp³)-H to C-N bond conversions.
- Previous Ritter-type C-H amination references should be added. (e.g., J. Am. Chem. Soc. 2012, 134, 5, 2547–2550; Chem. Sci., 2017, 8, 7180–7185; Chem. Commun. 2016, 52, 13082–13085) The difference between those methods should be clarified.
- Scheme 1, there should not be a comma between H₂SO₄ and its equivalence. Also, please draw the product derived from ethers and alcohols.
- Page 3 line 3, We speculated that SO₄⁻ might even oxidize the unactivated C-H bond to obtain carbocation. This is not appropriate statement since similar reactivity have been reported before (Org. Lett. 2016, 18, 1234–1237; Org. Lett. 2017, 19, 572–575). The author should do more literature research and add more important references.
- Table 1 entry 7-9, please organize it in the order ranging from low loading to high loading of sulfuric acid.
- Table 1 entry 6 and 10-13, please define the abbreviations in the caption.
- Table 1 entry 17, it is not clear whether the reaction is run with or without electricity. Under electrochemical condition, persulfate can be decomposed via electrochemical reduction which might explain the low yield. Please double check the yield when persulfate is used as oxidant under photochemical conditions and compare it with the standard photoelectrochemical conditions. This will justify the role of electricity.
- Figure 1B mechanism b, why does the carbocation localize on terminal position? It should be more stable if it locates at benzylic position.

- The formation of product 3 (derived from 1-bromoadamantane) and product 63-65 is interesting but it cannot be well explained by current hydrogen abstraction mechanism. Does the author have any other explanations on this?
- Scheme 3 product 62, the structure does not match the one in SI.
- In all figure caption, it is recommended to use XX as substrate to replace the product of XX which make it easier to read.

Overall, a major revision is recommended before publication.

Response to Reviewers

We are highly appreciating you for giving us an opportunity to revise the manuscript. We made the response, and modified the manuscript according to comments from three reviewers.

To Reviewer 1

The authors thanks very much for your valuable comments on our paper.

According to your comments, we made the following the response and modifications.

Major experimental revisions:

According to your comment 1: As authors note the reaction does proceed efficiently without light (Table 1 entries 1 (97%) vs 4 (74%), and Fig. 1A) and homolysis of persulfate is known at temperatures above rt (e.g. 40-60 °C). Elsewhere, increased reaction temperature as a result of hi power LED irradiation is known to promote photochemical reaction yields: ChemPhotoChem 2021, DOI:10.1002/cptc.202100059. Therefore, can this ca. +20% yield over 12-20 h rxn time simply be a thermal effect from LED heat on the homolysis of persulfate? (Same question over signal formation in EPR with LED). Of course the answer to this key above question influence the whole manuscript, including title. Authors must conduct reaction at a controlled, low temperature (e.g. 0 °C) to check the difference with vs without light. e.g. SI entry 8 at 10 °C (81%) should be checked without light. Authors should measure the internal temperature of a reaction under optimal conditions.

Response: Thank you for your nice comments.

We conducted control reactions without light several times under room temperature and 0 °C, and found light has no effect on the reaction.(Scheme R1) Under the

irradiation of white light, the internal temperature of a reaction increased to 30 °C from 0 °C.

The formation of sulfate radical anion might through the direct anodic oxidation of sulfate and the homolysis of persulfate generated by the dimerization of sulfate radical anion. Cathodic reduction of persulfate can replace light to provide sulfate radical anion. It may explain why light has no effect on the reaction. Besides, the water content has a great influence on the reaction. 74% conversion of Table 1 entry 4 was caused by the use of long-standing sulfuric acid with higher water content. 95-98% sulfuric acid is recommended to use for the electrolysis.

(1) Control experiments

(2) The generation of sulfate radical anion

Scheme R1. Control experiments

According to your comment 2: Persulfate is a white solid and does not absorb visible light in MeCN; its absorption is <400 nm.

It is possible a charge transfer complex between substrate and persulfate may operate to afford a visible absorption. Authors should report the UV-vis spectra of their reaction mixtures and components.

Response: Thank you for your nice comments.

As shown in figure below, persulfate does not absorb visible light in MeCN. The UV-vis spectra of the reaction mixtures of persulfate and 1,3-dimethyl-adamantane, persulfate and 4-ethylbenzonitrile, revealed that no absorption was appeared. The charge transfer complex between substrate and persulfate may not form under our conditions. We also conducted control reactions without blue light several times and found light has no effect on the reaction. It also proved that light had no effect on the electrolysis.

According to your comment 3: The claim results in Scheme 4A are photoelectrochemical is certainly a red flag. There is no obvious driving force for that C-O cleavage - how does the mechanism work in these cases? Classic acid-promoted SN1 Ritter reaction of activated alcohols using sulfuric acid is known even at rt, and cannot be ruled out. The reaction of tert-butanol is a textbook reaction students learn for the SN1 Ritter hydrolysis. Hydrolysis of ethers is also known to be achieved with strong acids.

Response: Thank you for your nice comments.

The mechanism proposed in the original manuscript cannot explain the C-O cleavage. Control experiments proved that electrolysis for the C-O cleavage is not necessary. The products derived from the C-O cleavage were omitted in the revised manuscript.

According to your comment 4: Authors must examine the control reactions (i.e. no light, no current, without both light+current) with adamantanol to check the efficiency. If it is confirmed to be simply a classic Ritter reaction, Scheme 4A's results can be omitted and this does not detract impact or hinder potential acceptance, in my view.

Response: Thank you for your nice comments.

As shown in Scheme R2, the Ritter reactions of adamantanol was performed with or without current and shown no difference of reaction outcomes between our method and the classic Ritter reaction. So, Scheme 4A's results were omitted in the revised manuscript.

Scheme R2. Ritter reaction of alcohol

According to your comment 5: Scheme 4A: Authors should check the reactions of 2x 4-substituted phenols (1x Elec-withdrawing group and 1x Elec-donating group), to

check if their C-O cleavage reactivity works for C(sp²)-OH.

Response: Thank you for your nice comments.

The electrolysis reactions of 4-isopropylphenol and 4-bromophenol were conducted and no product of C(sp²)-OH cleavage was detected. Our method does not work for the C(sp²)-OH cleavage.

Major text revisions:

According to your comment 1: sp² C-H amination is also reported in scheme 3. However, key related work on photoelectrochemical sp² C-H aminations, especially those of electron-deficient arenes, are missing in the citations and must be cited: *Angew. Chem. Int. Ed.* 2021, 60, 11163; *Org. Chem. Front.* 2021, 8, 1132; ChemRxiv 2021, DOI: 10.26434/chemrxiv.13718392.v1; *ChemElectroChem* 2021, 8, 1571. The authors's ms shows a key advantage over these reports in terms of simpler reaction conditions and shorter reaction times so this is useful for readers.

Response: Thank you for your nice comments.

As the mechanism of sp² C-H amination is different with sp³ C-H amination, the examples of **63-65** were removed in the revised manuscript. The key related works on photoelectrochemical sp² C-H aminations were also not cited as well. The suggested references have been cited properly.

According to your comment 2: Xu group recently reported a HCl enabled photoelectrochemical Minisci-type reaction. Anodic generated Cl₂ undergoes homolysis with vis light irradiation affording chlorine radicals which are powerful HAT agents that can activate also unactivated hydrocarbons (Angew. Chem. Int. Ed. 2020, 59, 14275). This seminal work, which also uses photoelectrochemistry with a simple mineral acid, is very related in concept to the current manuscript in mechanism. It is unfortunately not cited - it must be cited and described clearly in the introduction.

Response: Thank you for your nice comments.

In the revised manuscript, the seminal work reported by Xu group was cited and described clearly in the third paragraph.

According to your comment 3: Related to Scheme 4A: Authors should mention in the text that photoelectrochemical C-O (sp³/sp²) cleavage reactions are previously reported, but so far only under reductive conditions. These reports should be cited: Angew. Chem. Int. Ed. 2021, DOI:10.1002/anie.202105895 and Angew. Chem. Int. Ed. 2021, DOI: 10.1002/anie.202107169.

Response: Thank you for your nice comments.

In order to keep the consistency with the mechanism, the examples of C-O cleavage reactions were removed in the revised manuscript. Reviewer 2 also recommended us to remove the examples that are not irrelevant to C(sp³)-H amination. The key related works on photoelectrochemical sp² C-H aminations were also not cited as well.

Minor revisions:

According to your comment 1: The authors mention "Up to now, in the field of photoelectrochemistry, the conversion of C(sp³)-H bonds to C-N bonds is usually

constructed by coupling reaction of intramolecular amination." This is confusing, since up to now the conversion of C(sp³)-H bonds to C-N bonds are only by oxidation and then Ritter-type reaction (Ref 17) or hydrogen atom transfer (JACS 2020, 142, 1698 and JACS 2020, 142, 17693). These references should be cited and the description rephrased. This reviewer agrees that Refs 15-16 contain some related works, but many of them either i) not using photoelectrochemical methods or ii) are not C(sp³)-H aminations and they should be omitted or cited elsewhere in the manuscript.

Response: Thank you for your nice comments.

In the revised manuscript, the introduction was rewritten and focused on the C(sp³)-H amination by hydrogen atom transfer process. Reference 15 in the original manuscript were not relevant to the C(sp³)-H amination and deleted. References **16c** and **16e** in the original manuscript were deleted. References **16a**, **16b**, **16d**, **16f** were cited as references **27-38** in the revised manuscript.

According to your comment 2: Authors describe oxidative ability of sulfate radical, but sulfate radical is also a well-known HAT agent (Refs 19-21). Since the mechanism of unactivated Csp³-H bonds involves a HAT process, photocatalytic/photochemical persulfate/sulfate based HAT reactions should be briefly mentioned and relevant examples cited: Chem. Sci. 2016, 7, 2111-2118; Org. Lett. 2021, 23, 2002.

Response: Thank you for your nice comments.

In the revised manuscript, persulfate/sulfate based HAT reactions for Csp³-H functionalization by photocatalytic/photochemical strategy were briefly mentioned and cited (Chem. Sci. 2016, 7, 2111-2118; Org. Lett. 2021, 23, 2002).

According to your comment 3: Xu group recently reported a HCl enabled photoelectrochemical Minisci-type reaction. Anodic generated Cl₂ undergoes homolysis with visible light irradiation affording chlorine radicals which are powerful HAT agents that can activate also unactivated hydrocarbons (Angew. Chem. Int. Ed. 2020, 59, 14275). This seminal work, which also uses photoelectrochemistry with a

simple mineral acid, is very related in concept to the current manuscript in mechanism. It is unfortunately not cited - it must be cited and described clearly in the introduction.

Response: Thank you for your nice comments.

In the revised manuscript, the seminal work reported by Xu group was cited and described clearly in the third paragraph.

According to your comment 4: Related to Scheme 4A: Authors should mention in the text that photoelectrochemical C-O (sp³/sp²) cleavage reactions are previously reported, but so far only under reductive conditions. These reports should be cited: *Angew. Chem. Int. Ed.* 2021, DOI:10.1002/anie.202105895 and *Angew. Chem. Int. Ed.* 2021, DOI: 10.1002/anie.202107169.

Response: Thank you for your nice comments.

In order to keep the consistency with the mechanism, the examples of C-O cleavage reactions (Scheme 4A) were removed in the revised manuscript. The key related works on photoelectrochemical C-O (sp³/sp²) cleavage were also not cited as well. Related references have been cited.

According to your comment 5: Product 71 should be in Scheme 2, since the starting material is not alcohol or ether.

Response: Thank you for your nice comments.

Cyclohexanone went through aldol condensation and Ritter type reaction to form **71**. So, product **71** was removed as well.

According to your comment 6: pg4: "equivalents" should be equivalents.

Response: Thank you for your nice comments.

In pg4, "equivuivalents" was corrected to equivalents.

According to your comment 7: Table 1: conversion is shown instead of yield. Conversion only means how much starting material is consumed. Authors should show product yield.

Response: Thank you for your nice comments.

In Table 1, the outcome of the electrolysis was shown by yield instead of conversion.

According to your comment 8: Ref. 21's author names are incorrect, first names are given full and second name initials. It should be the other way around: Davide, R.  Ravelli, D. etc

Response: Thank you for your nice comments.

The author names in the ref were rechecked and modified.

According to your comment 9: Table 1: please define "CC" and "CF" (entries 11-12) since these electrodematerials are currently unclear

Response: Thank you for your nice comments.

The full names of "RVC" and "CF" were added in the footnote of Table 1.

According to your comment 10: Fig. 1: "shed light" : advise to rephrase this to avoid repetition of "light" and possible misunderstandings:

e.g. "in order to unravel/probe the reaction mechanism.."

Response: Thank you for your nice comments.

In the revised manuscript, "shed light" was rephrased to "obtain more insights".

According to your comment 11: Abstract+pg.2+throughout: Homogenised (scientifically incorrect, different meaning)  homolysed (correct)

Response: Thank you for your nice comments.

In the full revised manuscript, " homogenised " was corrected to " homolysed".

According to your comment 12: "deprived a H atom" (could be confused with deprotonation)  abstracted a H atom/underwent hydrogen atom transfer/HAT (clearer)

Response: Thank you for your nice comments.

In the full revised manuscript, "deprived a H atom" was corrected to " abstracted a H atom ".

According to your comment 13: Scheme 4b: show expanded structures of 72-78 to improve readability.

Response: Thank you for your nice comments.

In the revised manuscript, products **63-68** were shown as expanded structures.

According to your comment 14: Scheme 3: 49 vs 50, 58, 59-62: different depictions of ester structures. Please use structure 49's representation for all.

Response: Thank you for your nice comments.

According to your comment 15, some similar substrates were grouped by one drawn structure with an R group to simplify. "COOCH₃" was used to depict all ester structures.

According to your comment 15: Some substrates are similar and should be grouped by one drawn structure with an R group to simplify, e.g. 31 vs 49 (almost same), 34 vs 36 and 21 vs 22. Readers will not get value from seeing all the different R group structures drawn individually.

Response: Thank you for your nice comments.

In the scheme 3, some similar substrates were grouped by one drawn structure with an

R group to simplify.

To Reviewer 2

The authors thanks very much for your valuable comments on our paper.

According to your comments, we made the following the response and modifications.

According to your comment 1: Introduction: the Authors focus the attention on the functionalization of C-H bonds, but then their examples include the functionalization of different bond types (C-C via decarboxylation and C-halogen via dehalogenation, C-O bonds in alcohols/ethers). This inconsistency is rather confusing and should be addressed. I would personally remove examples not dealing with C-H bonds, also to be consistent with the mechanism. Furthermore, there is a huge difference between aliphatic vs aromatic C-H bonds (see the case of benzene) in terms of mechanism of activation (aromatic C-H bonds cannot be activated via C-H cleavage!). What about the use of ethers/alcohols? These are not even mentioned, neither in the Introduction, nor in the mechanism.

Response: Thank you for your nice comments.

The activation mechanism of C(sp³)-H bonds, C(sp²)-H bonds and C-X bonds (X= Br, COOH) are hydrogen atom transfer, single electron transfer, anodic reduction debromination and anodic oxidation decarboxylation respectively. In order to keep the consistency of mechanism, title and content, the products derived from the cleavage of C(sp²)-H bonds and C-X bonds (X= Br, COOH, OH, OR) were removed.

According to your comment 2: Introduction/2: a suitable scheme describing

literature precedents is lacking.

Response: Thank you for your nice comments.

The related elegant works focused on the HAT process for C(sp³)-H bond amination in electrochemistry and photoelectrochemistry were added in Scheme 1.

According to your comment 3: Chemistry: I am firmly convinced that the claim by the Authors, in particular the actual "photoelectrochemical" nature of their protocol, is not justified by their experimental data. Indeed, the data gathered in Table 1 (see entry 4) clearly demonstrate that light is not essential for the process to occur. 74% product yield cannot be considered a minor pathway and the system works nicely also under purely electrochemical conditions. This aspect is not even mentioned by the Authors!

Response: Thank you for your nice comments.

We conducted control reactions without blue light several times and found light has no effect on the reaction. Sulfate radical anion might form through the direct anodic oxidation of sulfate and the homolysis of persulfate generated by the dimerization of sulfate radical anion. Cathodic reduction of persulfate can replace light to provide sulfate radical anion. It may explain why light has no effect on the reaction. Besides, the water content has a great influence on the reaction. 74% conversion of Table 1 entry 4 was caused by the using of long-standing sulfuric acid with higher water content. 95-98% sulfuric acid is recommended to be used for the electrolysis.

According to your comment 4: Figure 1B: equation a in part B should be justified by experimental evidences, not only by literature precedents. On the other hand, equation b in part B is rather confusing because the Authors should actually justify

their claims:

* light: is persulfate absorbing light in the blue region?

* electricity: this activation manifold cannot lead to the formation of 2 equiv of sulfate radical anion;

* heat: the process occurs at 25°C, is persulfate homolyzed (not homogenized) at this temperature?

Also, the shown interaction of the persulfate radical anion with the substrate is not representative at all of the diverse interactions reported in the text (see the sub  sub.+ conversion).

Response: Thank you for your nice comments.

The UV-vis spectra of $K_2S_2O_8$ in CH_3CN revealed no absorption was afforded in the blue region. The cathodic reduction of persulfate provided one sulfate radical anion and one sulfate, not two sulfate radical anion. Persulfate could not be homolyzed to sulfate radical anion at 25°C.

According to your comment 5: Figure 1C: The Authors should better comment on the shape of the observed EPR signal. It seems that more than a single contribution is present. Also, the parameters of the signal are not consistent with the literature, see

DOI: 10.3109/10715769209083142.

Response: Thank you for your nice comments.

The EPR signal in Figure 1C is not matched with reported EPR spectra of DMPO-SO₄⁻. Currently, the attribution of the EPR signals has not been determined. In addition, with the exclusion of light effect on the electrolysis, the EPR experiment was also deleted in the revised manuscript.

To Reviewer 3

The authors thanks very much for your valuable comments on our paper.

According to your comments, we made the following the response and modifications.

According to your comment 1: The term photoelectrochemistry is misused. Photoelectrochemistry is commonly used to describe the use of photoelectrodes or photovoltaics for electrolysis. This work falls into the range of electrophotocchemistry. This definition is made in reference 8 of the manuscript.

Response: Thank you for your nice comments.

We conducted control reactions without blue light several times and found light has no effect on the rection. Sulfate radical anion might form through the direct anodic oxidation of sulfate and the homolysis of persulfate generated by the dimerization of sulfate radical anion. Cathodic reduction of persulfate can replace light to provide sulfate radical anion. It may explain why light has no effect on the rection. Photoelectrochemistry is not used in the revised manuscript.

According to your comment 2: Have the authors done a control experiment using a persulfate salt as the stoichiometric oxidant under photoirradiation? This should also provide the desired product based on the proposed mechanism.

Response: Thank you for your nice comments.

The control experiments using potassium persulfate as the stoichiometric oxidant under photoirradiation or at 80°C were performed and did not provide desired product. It may prove that the oxidation of carbon radical to cation is initiated by anode oxidation instead of sulfate radical anion.

According to your comment 3: In abstract line 3, “Then $S_2O_8^{2-}$ was homogenized under light to give sulfate radical anion, which deprived hydrogen on the substrate to generate radical intermediate.” Does the author have a typo mistake here? It should be homolyzed rather than homogenized. Same mistake was found in page 9 and 10. In addition, the verb deprive is recommended to be replaced by abstract.

Response: Thank you for your nice comments.

In the full revised manuscript, "homogenised" was corrected to "homolysed", "deprived a H atom" was corrected to "abstracted a H atom".

According to your comment 4: In abstract line 5, remove with in featured with.

Response: Thank you for your nice comments.

In abstract line 5, "featured with" was corrected to "featured".

According to your comment 5: In abstract line 7, change challenge into challenging.

Response: Thank you for your nice comments.

In abstract line 5, " challenge" was corrected to " challenging".

According to your comment 6: Rewriting introduction is recommended. There are many methods to construct C-N bond. Some of the important reactions (reductive amination, Ullman coupling) has not been mentioned. It is recommended to focus on the C(sp³)-H to C-N bond conversions.

Response: Thank you for your nice comments.

In the revised manuscript, the introduction was rewritten and focused on the C(sp³)-H amination by hydrogen atom transfer process. Some of the important reactions, including reductive amination, Ullman coupling, were mentioned and cited.

According to your comment 7: Previous Ritter-type C-H amination references should be added. (e.g., J. Am. Chem. Soc. 2012, 134, 5, 2547–2550; Chem. Sci., 2017, 8, 7180–7185; Chem. Commun. 2016, 52, 13082–13085) The difference between those methods should be clarified.

Response: Thank you for your nice comments.

In the revised manuscript, previous works related to Ritter-type C(sp³)-H amination have been cited and the difference between those methods and our work has been also described.

According to your comment 8: Scheme 1, there should not be a comma between H₂SO₄ and its equivalence. Also, please draw the product derived from ethers and alcohols.

Response: Thank you for your nice comments.

In the revised manuscript, Scheme 1 was rewritten to present the different HAT

reagents used for the C(sp³)-H amination in photoelectrochemistry or electrochemistry. The examples of ethers and alcohols were deleted.

According to your comment 9: Page 3 line 3, we speculated that SO₄^{·-} might even oxidize the unactivated C-H bond to obtain carbocation. This is not appropriate statement since similar reactivity have been reported before (Org. Lett. 2016, 18, 1234–1237; Org. Lett. 2017, 19, 572–575). The author should do more literature research and add more important references.

Response: Thank you for your nice comments.

In the revised manuscript, the introduction was rewritten and the sentence of “we speculated that SO₄^{·-} might even oxidize the unactivated C-H bond to obtain carbocation” was deleted.

According to your comment 10: Table 1 entry 7-9, please organize it in the order ranging from low loading to high loading of sulfuric acid.

Response: Thank you for your nice comments.

In the revised manuscript, Table 1 entries 7-9 were reorganized in the order ranging from low loading to high loading of sulfuric acid.

According to your comment 11: Table 1 entry 6 and 10-13, please define the abbreviations in the caption.

Response: Thank you for your nice comments.

In Table 1, the abbreviations of “RVC” and “CC” were defined in the footnotes.

According to your comment 12: Table 1 entry 17, it is not clear whether the reaction is run with or without electricity. Under electrochemical condition, persulfate can be decomposed via electrochemical reduction which might explain the low yield. Please

double check the yield when persulfate is used as oxidant under photochemical conditions and compare it with the standard photoelectrochemical conditions. This will justify the role of electricity.

Response: Thank you for your nice comments.

Table 1 entry 17 was not clearly described and the reaction was performed without electricity, providing 7% conversion of 1,3-dimethyladamantane and trace product. Using $K_2S_2O_8$ instead of H_2SO_4 as oxidant, the control reactions under heating, photochemical and electrochemical conditions were shown in Figure 2. The results may prove that the oxidation of carbon radical to cation proceed by anode oxidation instead of sulfate radical anion.

According to your comment 13: Figure 1B mechanism b, why does the carbocation localize on terminal position? It should be more stable if it locates at benzylic position.

Response: Thank you for your nice comments.

Allyl cation derived from oxidation of **28a-I** is C3 resonance structure and would be apt to be attacked in the terminal position due to its less steric.

According to your comment 14: The formation of product 3 (derived from 1-bromoadamantane) and product 63-65 is interesting but it cannot be well explained by current hydrogen abstraction mechanism. Does the author have any other explanations on this?

Response: Thank you for your nice comments.

As shown in Scheme R3, 1-bromoadamantane proceeds cathodic reduction to form bromide anion and carbon radical followed by anodic oxidation to give carbon cation successively. For benzene substrates, the electrochemical generated sulfate radical anion is served as SET agent rather than HAT agent. In order to keep the consistency with the mechanism, the examples 1-bromoadamantane and benzene substrates were removed in the revised manuscript.

Scheme R3. The electrolysis of 1-bromoadamantane and benzene.

According to your comment 15: Scheme 3 product 62, the structure does not match the one in SI.

Response: Thank you for your nice comments.

The structure of product 62 in the original manuscript has been corrected in the revised manuscript.

According to your comment 16: In all figure caption, it is recommended to use XX as substrate to replace the product of XX which make it easier to read.

Response: Thank you for your nice comments.

In the revised manuscript, the products derived from the cleavage of C-X bonds (X= Br, COOH, OH, OR) were removed. The figure caption was also rewritten.

REVIEWER COMMENTS

Reviewer #1 (Remarks to the Author):

The revised manuscript of Ye, Ma and co-workers reports a Ritter-type C-H amination reaction using sulfate anions as HAT agents. The authors are commended for carefully addressing the earlier reviewers' comments, including their thorough probing of the influence of light – which in the end was found to be unnecessary and promoted the reaction by simple thermal heat transfer. The manuscript has hence been restructured to focus only on the electrochemical transformation. I agree with the decision to remove some transformations within the manuscript (C-O cleavage, azolation). The authors' deserve respect for their transparency in removing the C-O cleavage which was found to be a classic Ritter reaction (i.e. no current gave same result). After looking through the revised manuscript and reevaluating its impact: The authors' method is able to ungate unactivated C-H bonds (not only benzylic) in synthetically useful aminations. Compared to previous reports (See comment below) the simplicity of conditions using H₂SO₄ as a source of sulfate radical instead of persulfate or more expensive SelectFluor/hypervalent iodine is indeed noteworthy, attractive and makes the chemistry general and user-friendly. Overall, I support its acceptance in Nat. Commun., but only after following revisions. Particularly, the manuscript needs to do better at citing and depicting previous works:

- While such C-H functionalization/amination reactions have been reported before using other electrochemical and ground-state chemical methods (e.g. Ref. 58, 60, 46, 41), the depiction of previous literature in Fig. 1 is (while interesting) not directly related to the author's transformation. There they show azidations and azolations, but Ritter-type chemistry where formal addition of MeCN occurred is now shown - this gives an impression of an over-inflated impact. While all these cited works should absolutely be maintained, it is more helpful to the reader when the authors give due visibility to the previous methods that actually reflect their specific Ritter-type transformation. Authors should split Figure 1 into parts to compare (i) ground state chemical methods (F-TEDA-PF₆, hypervalent iodine: Refs 58,60, tert-butylhydroperoxide which was reported by Hartwig and surprisingly not mentioned: JACS 2014, 136, 2555)

and (ii) (electro)photocatalytic methods (TAC⁺, Cu catalysis/tBuOOH) Ref. 41, and the following important papers that were missed off: Science 2021, 371, 620; and ACS Catal. 2020, 10, 8582.

To highlight the importance of their work, I suggest to authors to emphasize their simpler reaction conditions. In ground-state chemistry reports, the HAT reagents SelectFluor and hypervalent iodine are expensive. In the photoelectrochemical reports, limitation is always for benzylic C-H bond Ritter amination. Yields of reactions using tBuOOH HAT (JACS[^] and ACS Catal[^]) are typically lower.

- A HAT activation of unactivated hydrocarbon C-H bonds recently appeared and should be cited: Chem. Comm. 2021, 57, 4424 but it does not need to be mentioned in the Fig. 1

- In Fig. 1, half of the examples mentioned are electrophotocatalytic processes, and authors later mention a photoelectrochemical activation of unactivated C-H bonds by chlorine radicals of Xu and co-workers. photoelectrochemistry is not yet a mature research field, so authors should refer readers unfamiliar to the field to some introductory reviews ACIE 2020, 59, 11732-11747; OCF 2020, 7, 131-135 and reviews that cover its applications in HAT/C-N bond formations: Chem. Soc. Rev. 2021, DOI: 10.1039/D1CS00223F ; ACIE 2021, DOI: 10.1002/anie.202107811. Appropriate reviews are already given for electrochemical HAT.

Elsewhere, enough detail is provided to reproduce the work, the data analysis is sound and the claims and conclusions made by authors are reasoned well and supported by their control and mechanistic experiments.

Reviewer #4 (Remarks to the Author):

The authors of this manuscript present an electrochemical method, which can be used to functionalize certain (such as adamantane) alkanes via C-H activation by a Ritter-type reaction. I was asked to be a referee for this paper on its second submission to Nature Communications. Specifically, I was supposed to check, whether the issues raised by referees 2 and 3 were addressed.

The comments from the referees 2 and 3 mainly dealt with issues with the former mechanistic proposal – as the paper was originally submitted as a photo-electro catalysis paper. Thanks to the comments of the reviewers, it was however discovered that the process does efficiently proceed in the absence of light, and should be regarded as a regular electrosynthetic (electrooxidative) reaction. As such, most of the comments of the referees 2 and 3 are no longer relevant in this manuscript, as the proposed mechanism was changed drastically in comparison with the original submission.

I have to say, that according to my opinion, the proposed mechanism is still somewhat unclear. As the active species, which initiates the reaction by hydrogen atom transfer, $\text{SO}_4^{\cdot-}$ radical is proposed – supposedly obtained by an oxidation of the sulfate anion on the electrode. The required potential to achieve oxidation of sulfate is very high – and it is likely that other species present in the reaction

system would get oxidized at such potential instead. The authors should try to show, that the sulfate really gets oxidized under their conditions. Running cyclic voltammetry experiments with all of the species present in the system (adamantane, sulfuric acid, and MeCN/electrolyte as blank) would be a good start. If the sulfuric acid would indeed be the first species oxidized, then another experiment should be performed – cyclic voltammetry of sulfuric acid with increasing concentrations of adamantane – if adamantane is oxidized by an active species generated by the oxidation of sulfuric acid, then a catalytic wave should be present in the CV.

My main issue with the manuscript is the existence of prior work from 1970s (uncited in the manuscript), which propose a direct electrooxidation of adamantane-type systems, and their Ritter reaction with acetonitrile. See *Tetrahedron Lett* 1976 631-634 (10.1016/S0040-4039(00)77931-7) and *J Chem Soc Perkin I* 1977 1831-1834 (10.1039/P19770001831). If the main reaction pathway in this manuscript is also via the direct oxidation of the adamantane, and the sulfate acts just as an additive that helps with the Ritter-step (as it is in *Russ J Org Chem* 2017, 1170-1175), then the authors have simply re-discovered an already known reaction from the 1970s.

Minor points: The quality of the sulfuric acid seems to be critical for this reaction. Yet nowhere in the manuscript not SI was the type (even concentration!), manufacturer, etc. of the acid used disclosed.

Manuscript contains some awkward writing and grammar errors (e.g. page 2: "In particular, photoredox generated excited state of decatungstate anion (W; [W10O32]4-)19, nitrogen radical20-22, oxygen radical23-25 and bromine radical26 have employed as the efficient HAT agents for the selective amination of C(sp3)-H bonds.") Manuscript should be carefully proof-read prior to its publication.

This issue is even worse in the SI (e.g. page 3 of the SI: "Wash the extracted organic phase with water (10 mL) and saturated brine (10 mL), then dried by Na2SO4. 87") Please check the SI thoroughly, and correct the English and typos.

The 13C NMR of compound 42 is not reported correctly (splitting with fluorine). Please correct.

As a result, I believe that I cannot recommend this manuscript for publication in a high-impact journal such as *Nature Communications*. In the current stage, this work seems to be a rediscovery of an electrooxidation already reported in the 70s.

Response to Reviewers

We are highly appreciating you for giving us an opportunity to revise the manuscript. We made the response, and modified the manuscript according to comments from two reviewers.

To Reviewer 1

The authors thanks very much for your valuable comments on our paper.

According to your comments, we made the following the response and modifications.

According to your comment 1: Particularly, the manuscript needs to do better at citing and depicting previous works: While such C-H functionalization/amination reactions have been reported before using other electrochemical and ground-state chemical methods (e.g. Ref. 58, 60, 46, 41), the depiction of previous literature in Fig. 1 is (while interesting) not directly related to the author's transformation. There they show azidations and azolations, but Ritter-type chemistry where formal addition of MeCN occurred is now shown - this gives an impression of an over-inflated impact. While all these cited works should absolutely be maintained, it is more helpful to the reader when the authors give due visibility to the previous methods that actually reflect their specific Ritter-type transformation. Authors should split Figure 1 into parts to compare (i) ground state chemical methods (F-TEDA-PF₆, hypervalent Iodine: Refs 58,60, tert-butylhydroperoxide which was reported by Hartwig and surprisingly not mentioned: JACS 2014, 136, 2555) and (ii) (electro)photocatalytic methods (TAC⁺, Cu catalysis/tBuOOH) Ref. 41, and the following important papers that were missed off: Science 2021, 371, 620; and ACS Catal. 2020, 10, 8582. To highlight the importance of their work, I suggest to authors to emphasize their simpler reaction conditions. In ground-state chemistry reports, the HAT reagents SelectFluor and hypervalent iodine are expensive. In the photoelectrochemical reports, limitation is always for benzylic C-

H bond Ritter amination. Yields of reactions using tBuOOH HAT (JACS[^] and ACS Catal[^]) are typically lower.

Response: Thank you for your comments.

In the revised manuscript, we have reorganized the content of Figure 1 and added the Ritter-type transformations. (*J. Am. Chem. Soc.* 2012, 134, 2547-2550; *Chem. Sci.* 2017, 8, 7180-7185; *J. Am. Chem. Soc.* 2021, 143, 8597-8602; *Science.* 2021, 371, 620–626.) Other related works (*J. Am. Chem. Soc.* 2014, 136, 2555–2563; *ACS Catal.* 2020, 10, 8582–8589.) were also depicted in Figure 1. These works for the amination of C(sp³)-H bonds in Figure 1 involved both ground state chemical methods and (photo)electrochemical methods. The simpler reaction conditions and broader substrates applicability of our method were also demonstrated.

According to your comment 2: A HAT activation of unactivated hydrocarbon C-H bonds recently appeared and should be cited: *Chem. Comm.* 2021, 57, 4424 but it does not need to be mentioned in the Fig. 1

Response: Thank you for your comments.

In the revised manuscript, the work recently reported by Ravelli (*Chem. Comm.* 2021, 57, 4424-4427.) has been cited as reference 22.

According to your comment 3: In Fig. 1, half of the examples mentioned are electrophotocatalytic processes, and authors later mention a photoelectrochemical activation of unactivated C-H bonds by chlorine radicals of Xu and co-workers. photoelectrochemistry is not yet a mature research field, so authors should refer readers unfamiliar to the field to some introductory reviews *ACIE* 2020, 59, 11732-11747; *OCF* 2020, 7, 131-135 and reviews that cover its applications in HAT/C-N bond formations: *Chem. Soc. Rev.* 2021, DOI: 10.1039/D1CS00223F ; *ACIE* 2021, DOI: 10.1002/anie.202107811. Appropriate reviews are already given for electrochemical HAT.

Response: Thank you for your comments.

The introductory reviews and reviews that cover its applications in HAT/C-N bond formations have been cited in the revised manuscript.

To Reviewer 4

The authors thanks very much for your valuable comments on our paper.

According to your comments, we made the following the response and modifications.

According to your comment 1: I have to say, that according to my opinion, the proposed mechanism is still somewhat unclear. As the active species, which initiates the reaction by hydrogen atom transfer, $\text{SO}_4\cdot^-$ radical is proposed – supposedly obtained by an oxidation of the sulfate anion on the electrode. The required potential to achieve oxidation of sulfate is very high – and it is likely that other species present in the reaction system would get oxidized at such potential instead. The authors should try to show, that the sulfate really gets oxidized under their conditions. Running cyclic voltammetry experiments with all of the species present in the system (adamantane, sulfuric acid, and MeCN/electrolyte as blank) would be a good start. If the sulfuric acid would indeed be the first species oxidized, then another experiment should be performed – cyclic voltammetry of sulfuric acid with increasing concentrations of adamantane – if adamantane is oxidized by an active species generated by the oxidation of sulfuric acid, then a catalytic wave should be present in the CV.

Response: Thank you for your comments.

We agree with the Reviewer I that cyclic voltammetry is less relevant anyway in constant current electrolytic reactions vs potentiostatic. The outcomes of the electrolysis of adamantane in constant current and constant potential are very different. For example, under 5 mA constant current, the direct electrolysis of 1,3-

dimethyladamantane with the absence of sulfuric acid only provided 5% yield of monoacetamidated product. 74% Yield of monoacetamidated product and 58% yield of diacetamidated product were afforded under the constant potential electrolysis of adamantane at 2.5V and 3.0V (vs Ag wire) respectively. (*Tetrahedron Lett*, 1976, 17, 631-634, doi 10.1016/S0040-4039(00)77931-7). Besides, literature values of oxidation potentials of sulfate, methanesulfonic acid, and some alkanes were reported. The oxidation potentials of these compounds are shown in the table 1. So, sulfates are likely to be oxidized by anode in prior to alkanes.

In the revised manuscript, the comparison of oxidation potentials between sulfate and some alkanes was added, the related literatures were also cited as well.

Table 1. The oxidation potentials of the substrate.

Substrate	Oxidation potential	Reference
SO ₄ ²⁻	$E^o=2.6$ V (vs NHE)	Adv. Phys. Org. Chem. , 1982, 18, 79-185; Chem. Commun. , 2013, 49, 7480-7482.
CH ₃ SO ₃ H	2.9-3.7 V (vs RHE)	Russ J Electrochem , 2019, 55, 579–589.
2-Methylpentane	$E_{1/2}=3.01$ V(vs Ag/Ag ⁺)	Tetrahedron Lett , 1968, 9, 6255-6258.
n -Heptane	$E_{1/2}>3.4$ V (vs Ag/Ag ⁺)	Tetrahedron Lett , 1968, 9, 6255-6258.
n -Hexane	$E_{1/2}>3.4$ V (vs Ag/Ag ⁺)	Tetrahedron Lett , 1968, 9, 6255-6258.
Adamantane	$E^o=3.01$ V (vs SCE)	Electrochimica Acta , 2009, 54, 5959–5963.
Norbornane	$E^o =2.95$ V (vs SCE)	Electrochimica Acta , 2009, 54, 5959–5963.

According to your comment 2: My main issue with the manuscript is the existence of prior work from 1970s (uncited in the manuscript), which propose a direct electrooxidation of adamantane-type systems, and their Ritter reaction with acetonitrile. See *Tetrahedron Lett* 1976 631-634 (10.1016/S0040-4039(00)77931-7) and *J Chem Soc Perkin I* 1977 1831-1834 (10.1039/P19770001831). If the main reaction pathway in this manuscript is also via the direct oxidation of the adamantane, and the sulfate acts

just as an additive that helps with the Ritter-step (as it is in Russ J Org Chem 2017, 1170-1175), then the authors have simply re-discovered an already known reaction from the 1970s.

Response: Thank you for your comments.

We have carefully considered the reviewer's concern about our work which is the existence of prior work from 1970s. However, we think our work is different from the prior work in 1970s, the reasons are as follows. First, control experiments indicated that sulfuric acid is the pivotal factor in our electrolysis system. Without sulfuric acid, only 5% yield of desired product was afforded. Replacing sulfuric acid with sodium sulfate, potassium peroxodisulfate and methanesulfonic acid, 76-89% yields were obtained. The electrolysis of methanesulfonic acid also leads to the formation of dimethyl disulfoperoxide via the dimerization of mesylate radicals. These alternatives to sulfuric acid are unlikely to facilitate the Ritter-step. In SI, we also proved the formation of persulfate by UV absorption spectra experiment. So, sulfuric acid does not act as an additive to facilitate the Ritter-step. Next, the value of the research work cannot be judged by the already known reaction of a certain substrate. The direct electrolysis of adamantane derivatives to prepare acetamidated product were reported by Mellor (*Tetrahedron Lett*, 1976, 17, 631-634; *J. Chem. Soc. Perkin. I.*, 1977, 16, 1831-1834.). And we all known that tertiary alkanes are more active and easily oxidized than secondary alkanes. The Ritter type amination of simple secondary alkanes, such as cyclopentane, cyclohexane, cycloheptane, cyclononane and *n*-hexane, via the direct electrolysis has not been reported. But, this type of alkanes are viable reactants under our electrolysis system. So, our method is not just the re-discovery of the already known reaction, but an important advancement of the older type reaction. Just like the work reported by Baran in 2016, they greatly improved the efficiency, selectivity and substrate scope of the already known anodized allylic C–H oxidation by changing the mediator from NHPI to Cl₄NHPI. (*Nature*, 2016, 533, 77–81; *Chem. Pharm. Bull.*, 1985, 33, 4798–4802.)

In the revised manuscript, the discussion about the role sulfate played in the electrolysis was added. The literature about the direct electrolysis of adamantane derivatives (*Tetrahedron Lett*, 1976, 17, 631-634; *J. Chem. Soc. Perkin. I.*, 1977, 16, 1831-1834.) were cited as reference 34-35.

According to your comment 3: Minor points: The quality of the sulfuric acid seems to be critical for this reaction. Yet nowhere in the manuscript not SI was the type (even concentration!), manufacturer, etc. of the acid used disclosed.

Response: Thank you for your comments.

The quality of the sulfuric acid is indeed critical for this reaction. When the water content of sulfuric acid increases, it is unfavorable for the reaction. In the revised manuscript, we have marked the concentration of sulfuric acid.

According to your comment 4: Manuscript contains some awkward writing and grammar errors (e.g. page 2: "In particular, photoredox generated excited state of decatungstate anion (W; $[W_{10}O_{32}]^{4-}$)¹⁹, nitrogen radical²⁰⁻²², oxygen radical²³⁻²⁵ and bromine radical²⁶ have employed as the efficient HAT agents for the selective amination of C(sp³)-H bonds.") Manuscript should be carefully proof-read prior to its publication. This issue is even worse in the SI (e.g. page 3 of the SI: "Wash the extracted organic phase with water (10 mL) and saturated brine (10 mL), then dried by Na₂SO₄. 87") Please check the SI thoroughly, and correct the English and typos.

Response: Thank you for your comments.

We have carefully checked the manuscript and corrected the grammar errors in the revised manuscript to improve manuscript quality. In the revised supporting information, we have also rewritten the experimental operation section.

According to your comment 5: The ¹³C NMR of compound 42 is not reported correctly (splitting with fluorine). Please correct.

Response: Thank you for your comments.

The ^{13}C NMR of compound 42 in the original manuscript has been corrected in the revised supporting information

Reviewer #1 (Remarks to the Author):

The authors have well-met the concerns raised by reviewers in the previous round of revisions. Fig.1 and the related literature is now better depicted.

I totally agree with Reviewer 4 bringing up the papers of Mellor (Refs 35-36), which authors have now cited and given good visibility. For me though, the synthetic advance in the current manuscript is absolutely clear: Mellor's papers are more conceptual discoveries, apply only to adamantane derivatives and yields are generally lower even for these. While CV was measured for other hydrocarbons, they were not subjected to Rittertype amination. In contrast, the manuscript of the authors herein boasts an impressive scope of applications including secondary hydrocarbons (compounds 9-19). The selectivity of which is strongly reminiscent of a HAT mechanism. A direct electrolysis activation of unactivated (even electron withdrawn) hydrocarbons could not feasibly provide such yields and selectivity patterns.

For me, the fact the same reactivity/yields are observed with other sources of SO_4 radicals (table 1 entries 14-15), where sulfuric is absent, is consistent with this and is not consistent with the notion of a sulfuric-promoted direct oxidation of hydrocarbons.

[On this note, authors should reflect on entry 16. Authors say this is 'equivalent' to peroxydisulfate but not so - The radical would presumably be CH_3SO_3 not SO_4 . Rephrasing is necessary e.g. the word 'similar']

I note that the 1976 paper contains no experimental so will be near-impossible for contemporary users to repeat - an unfortunate general issue for a lot of these classical direct electrolysis reports.

I do think that the addition of a cyclic voltammetry comparison suggested by Reviewer 4 would add value to the paper. Interestingly, the 1976 paper appears to show that in TFA as solvent the oxidation potential of adamantane might decrease in acid, but this is not conclusive since reference electrodes are not identical.

But although it is a 'nice to have' this CV experiment, I do not see this really necessary - authors have provided redox potentials to support their argument. Although redox potentials are again quoted vs different reference electrodes, it is convincing that SO_4^{2-} oxidation is (at least slightly) thermodynamically favoured vs. hydrocarbons. Naturally the constant current direct electrolysis process starts to depend more on mass transfer, part of which includes concentration gradients and diffusion coefficients for which I expect SO_4^{2-} to outcompete hydrocarbons.

I agree with Reviewer 4 that details should be provided on the quality of H₂SO₄. At the moment, we see 4eq. and "(w = 98%)". I think that providing the readers with an additional number for concentration of acid in MeCN (can be in the footnote of each optimization/scope table) is more useful and direct.

Ref.4 was right to flag the grammar/spelling/language issues of the manuscript/SI, which I advise the authors/Editorial team to work together on polishing prior to final publication. For example: Ref 63 "Euroapean"  European "Ritter-step"  Ritter-type step. It is not the role of the reviewer to be a detailed language proofreader, but for me, the science of the manuscript is clearly understandable even if the English is not perfect.

After addressing these minor comments, I fully support acceptance of the manuscript without further peer review.

Dear Reviewers:

We have carefully proofread the main text and reference parts, and the modified content was highlighted in the revised manuscript, including the writing of Chinese names, punctuation and spelling problems.

Best regards,

Jinxing Ye

Corresponding author

School of Biomedical and Pharmaceutical Sciences, Guangdong University of Technology, Guangzhou 510006, China

E-mail: jinxingye@gdut.edu.cn.

Engineering Research Center of Pharmaceutical Process Chemistry, Ministry of Education, School of Pharmacy, East China University of Science and Technology, 130 Meilong Road, Shanghai 200237, China.

E-mail: yejx@ecust.edu.cn